*Report*

EMBO
*reports*

# A transcription factor module mediating C$_2$ photosynthesis in the *Brassicaceae*

Patrick J Dickinson [1]✉, Sebastian Triesch [2], Urte Schlüter[2], Andreas P M Weber [2] & Julian M Hibberd [1]✉

## Abstract

C$_4$ photosynthesis has arisen from the ancestral C$_3$ state in over sixty lineages of angiosperms. It is widely accepted that an early step in C$_4$ evolution is associated with the appearance of so-called C$_2$ photosynthesis caused by loss of glycine decarboxylase activity from mesophyll cells followed by activation in the bundle sheath. Although changes in *cis* to a distal enhancer upstream of the P-subunit of *GLYCINE DECARBOXYLASE* (*GLDP*) from C$_2$ *Moricandia* enable loss of expression from mesophyll cells, the mechanism then allowing *GLDP* expression in the bundle sheath is not known. Here we identify a MYC-MYB transcription factor module previously associated with the control of glucosinolate biosynthesis as the basis of this foundational event in the evolution of C$_2$ photosynthesis. Specifically, we find that in the C$_3$ state this MYC-MYB module already patterns *GLDP* expression to bundle sheath cells. As a consequence, when *GLDP* expression is lost from the mesophyll, the MYC-MYB dependent expression in the bundle sheath is revealed. Evolution of C$_2$ photosynthesis is thus associated with a MYC-MYB based transcriptional network already present in the C$_3$ state. This work identifies a molecular genetic mechanism underlying the bundle sheath accumulation of glycine decarboxylase required for C$_2$ photosynthesis and thus a fundamental step in the evolution of C$_4$ photosynthesis.

**Keywords** C2 Photosynthesis; C3 Photosynthesis; C4 Photosynthesis; Glycine Decarboxylase P Subunit; Bundle Sheath
**Subject Categories** Metabolism; Organelles; Plant Biology

## Introduction

Fixation of CO$_2$ during photosynthesis is central to life. In plants this is dependent on Ribulose 1,5-Bisphosphate Carboxylase Oxygenase (RuBisCO) operating as part of the Calvin-Benson-Bassham cycle. However, in addition to reacting with CO$_2$ RuBisCO catalyses a side-reaction with O$_2$ to produce the toxic metabolite phosphoglycolate (Bowes et al, 1971). The photorespiratory pathway metabolises phosphoglycolate, but CO$_2$ is lost and ATP, NADPH and amino acids are required (Tolbert, 1971). As temperatures increase the ratio of oxygenation to carboxylation reactions at the RuBisCO active site increases and so losses from photorespiration become more significant (Jordan and Ogren, 1984). It is widely thought that carbon concentrating mechanisms such as C$_4$ photosynthesis evolved to reduce the metabolic costs of photorespiration. In the case of the C$_4$ pathway this involves modifications to leaf anatomy, cell biology and biochemistry (Hatch, 1987). Typically, after conversion of CO$_2$ to bicarbonate by carbonic anhydrases C$_4$ biochemistry enables fixation by the enzyme phospho*enol*pyruvate carboxylase in mesophyll cells. Subsequent decarboxylation of C$_4$ acids releases high concentrations of CO$_2$ in a compartment such as the bundle sheath (Sage, 2001; Christin et al, 2013) and so the oxygenase activity of RuBisCO is reduced (Leegood, 2002; Carmo-Silva et al, 2015).

Some genera contain species that possess anatomical and biochemical characteristics associated with both C$_3$ and C$_4$ photosynthesis. Such plants are known as C$_3$-C$_4$ intermediates and a subset of these with an active mesophyll to bundle sheath glycine shuttle are known as C$_2$ species (Sage et al, 2012; Lundgren, 2020). Although statistical modelling predicted that the order of C$_4$ trait acquisition is flexible (Williams et al, 2013), it appears that two key enabling events are an increase in bundle sheath to mesophyll ratio (Christin et al, 2013; Edwards, 2019), and then a shift of glycine decarboxylase away from mesophyll cells such that its activity is restricted to the bundle sheath (Rawsthorne et al, 1988; Morgan et al, 1993; Mallmann et al, 2014; Blätke and Bräutigam, 2019). Repositioning of glycine decarboxylase to the bundle sheath is conjectured to initiate greater rates of CO$_2$ release and thus increased photosynthetic activation of this tissue (Keerberg et al, 2014). The two-carbon glycine molecule thus provides CO$_2$ for photosynthesis in bundle sheath tissue and, as such, led to the term C$_2$ photosynthesis. Glycine decarboxylase is made up of four subunits and loss of expression of the P-subunit (*GLDP*) from the mesophyll has repeatedly driven the appearance of C$_2$ photosynthesis (Rawsthorne et al, 1988; Morgan et al, 1993; Schulze et al, 2016). One example of this is found in the Brassicaceae family where the *Moricandia* genus contains both C$_3$ and C$_2$ species (Schlüter et al, 2017; Schlüter et al, 2023).

In the Brassicaceae a region referred to as the mesophyll (M) box is highly conserved in promoters of the *GLDP1* gene from C$_3$ and C$_2$ species (Adwy et al, 2015). Promoter deletion analysis showed that this region is involved in driving expression in mesophyll cells in *A. thaliana* and C$_3$ *M. moricandioides*

[1]Department of Plant Sciences, University of Cambridge, Downing Street, Cambridge CB2 3EA, UK. [2]Institute of Biochemistry, Heinrich-Heine University, 40225 Düsseldorf, Germany. ✉E-mail: patrick.dickinson@bristol.ac.uk; jmh65@cam.ac.uk

(Adwy et al, 2015; Adwy et al, 2019). Insertion of transposable elements between the M box and the core promoter is thought to abolish mesophyll expression of *GLDP1* in $C_2$ species leading to bundle sheath preferential expression (Triesch et al, 2024). In contrast to our understanding of how loss of mesophyll expression of *GLDP1* is brought about, the molecular architecture enabling emergence of bundle sheath *GLDP1* expression in the Brassicaceae has not yet been defined. Using $C_3$ *Arabidopsis thaliana* we first show that a bipartite MYC and MYB transcription factor module responsible for directing the transcription factor *MYB76* and thus glucosinolate biosynthesis genes to the bundle sheath is also able to pattern the *GLDP1* gene to this tissue. In the $C_3$ state this MYC-MYB module therefore operates in parallel with the M box (Adwy et al 2015) to ensure expression in both mesophyll and bundle sheath cells. The MYC-MYB binding sites are conserved in $C_2$ *M. arvensis* whereas insertion of transposable elements has shifted the M-box to disrupt function (Triesch et al, 2024). Therefore, this MYC-MYB module allows expression of *GLDP1* and assembly of the glycine decarboxylase holoprotein specifically in bundle sheath cells. We thus identify a molecular architecture in the $C_3$ state operating in both *cis* and *trans* that underpins a foundational trait associated with the evolution of $C_2$ and $C_4$ photosynthesis.

## Results and discussion

### In $C_3$ *A. thaliana* MYC and MYB transcription factors drive expression in the bundle sheath which combined with a mesophyll module generates broad expression across the leaf

*A. thaliana* contains two copies of *GLDP*, both of which are expressed in leaves (Appendix Fig. S1A) (Aubry et al, 2014). However, all Brassicaceae lineages containing $C_2$ species belong to the monophyletic Brassiceae tribe that has lost *GLDP2* such that *GLDP1* is the only remaining copy of the gene (Schlüter et al, 2017). To better understand the molecular basis of $C_2$ photosynthesis, we therefore focused on understanding the expression of *GLDP1*. As expected, the *GLDP1* promoter from $C_3$ *A. thaliana* drove constitutive expression in leaves, with expression in mesophyll cells (Fig. 1A; Appendix Fig. S2) and bundle sheath strands (Appendix Fig. S2). And consistent with previous analysis (Adwy et al, 2015), a 5' deletion removing the M box revealed that a proximal region comprising 561 nucleotides was sufficient to generate expression in bundle sheath strands (Fig. 1B; Appendix Fig. S3).

To define mechanisms controlling this cryptic expression of *GLDP1* in bundle sheath strands of *A. thaliana* we first used publicly available data to identify potential transcription factor binding sites in the promoter (Fig. 1C). Because not all binding sites have been defined and some transcription factors are predicted to bind to the same or very similar sequences, we clustered motifs from the JASPAR database (Fornes, 2020) by similarity (Dataset EV1). This provided an indication of transcription factor families likely able to bind the *GLDP1* promoter. In the 59 base pair M box, binding sites for C2H2, MADS, bZIP and BPC transcription factor families were present (Fig. 1C) suggesting that members of these families could be involved in generating mesophyll expression. Previous work had shown that nucleotides -561 to -295 upstream of

the translational start site of *GLDP1* are necessary for expression in bundle sheath strands (Adwy et al, 2015) however, it is not known if they are sufficient for this patterning. We therefore searched for transcription factor binding sites in this region but also in sequence up to the translational start site (Fig. 1C). Motifs associated with sixteen families of transcription factor families were identified, and this included closely spaced MYELOCYTOMATOSIS (MYC—belonging to the bHLH family) and MYOBLASTOMA (MYB) binding sites (Fig. 1C).

A bipartite module involving MYC2,3&4 and MYB28&29 directs expression of the *MYB76* transcription factor and glucosinolate biosynthesis genes to the bundle sheath of *A. thaliana* (Dickinson et al, 2020). Re-analysis of publicly available data showed that *GLDP1* expression was not reduced in leaves of the triple *myc2/3/4* mutant (Major et al, 2017). However, in the double *myb28/29* mutant (Burow et al, 2015) a small reduction in *GLDP1* transcript abundance was apparent (Appendix Fig. S1B). As the bundle sheath of *A. thaliana* comprises only ~15% of all cells in the leaf (Kinsman and Pyke, 1998) expression of *GLDP1* in the mesophyll will dominate signal from whole leaves. We thus consider this small change in *GLDP1* expression in the *myb28/29* double mutant consistent with MYB transcription factors controlling *GLDP1* expression in the bundle sheath. We hypothesised that the closely spaced MYC and MYB motifs drive expression of *AtGLDP1* in bundle sheath strands that becomes easily detectable once function of the mesophyll box is lost. Consistent with this, although they had not identified the MYC binding site, Adwy et al (2015) reported that loss of nucleotides −305 to −299 containing this sequence abolished expression in bundle sheath strands. The importance of the MYB site between nucleotides −284 and −277 (Fig. 1C) was not investigated. We therefore conjectured that the region containing only the MYC binding site would not be sufficient for expression in bundle sheath strands. Consistent with this, when nucleotides −561 to −295 were fused to the minimal CaMV35S promoter, GUS was not detected in leaves (Fig. 1D; Appendix Fig. S4). In contrast, when nucleotides −561 to −247, which also include the MYC and MYB sites, were fused to the minimal CaMV35S promoter, bundle sheath strand expression was restored (Fig. 1E; Appendix Fig. S5). Sequence downstream of the MYB binding site is therefore not necessary for expression in bundle sheath strands. To test whether sequence upstream of the MYC binding site is necessary for expression nucleotides from −347 to the ATG were fused to GUS. This construct drove expression in bundle sheath strands (Fig. 1F; Appendix Fig. S6). We conclude that nucleotides upstream of the MYC and MYB sites are not necessary for bundle sheath strand expression, and therefore that an enhancer positioned between nucleotides −305 to −277 upstream of the $C_3$ *A. thaliana* GDLP1 gene containing closely spaced MYC and MYB binding sites is responsible for generating bundle sheath expression. More broadly, this indicates that the MYC-MYB module can act alone to generate expression of genes such as *MYB76* in bundle sheath strands (Dickinson et al, 2020), but as seen for *AtGLDP1*, it can also act in concert with other elements such as the M box to ensure expression in both bundle sheath and mesophyll cells. This is consistent with work showing that constitutive expression patterns are generated through different promoter regions generating expression in different cell types (Cai et al, 2020). We next sought to test whether the MYC-MYB module is conserved in *GLDP1* genes from $C_2$ species.

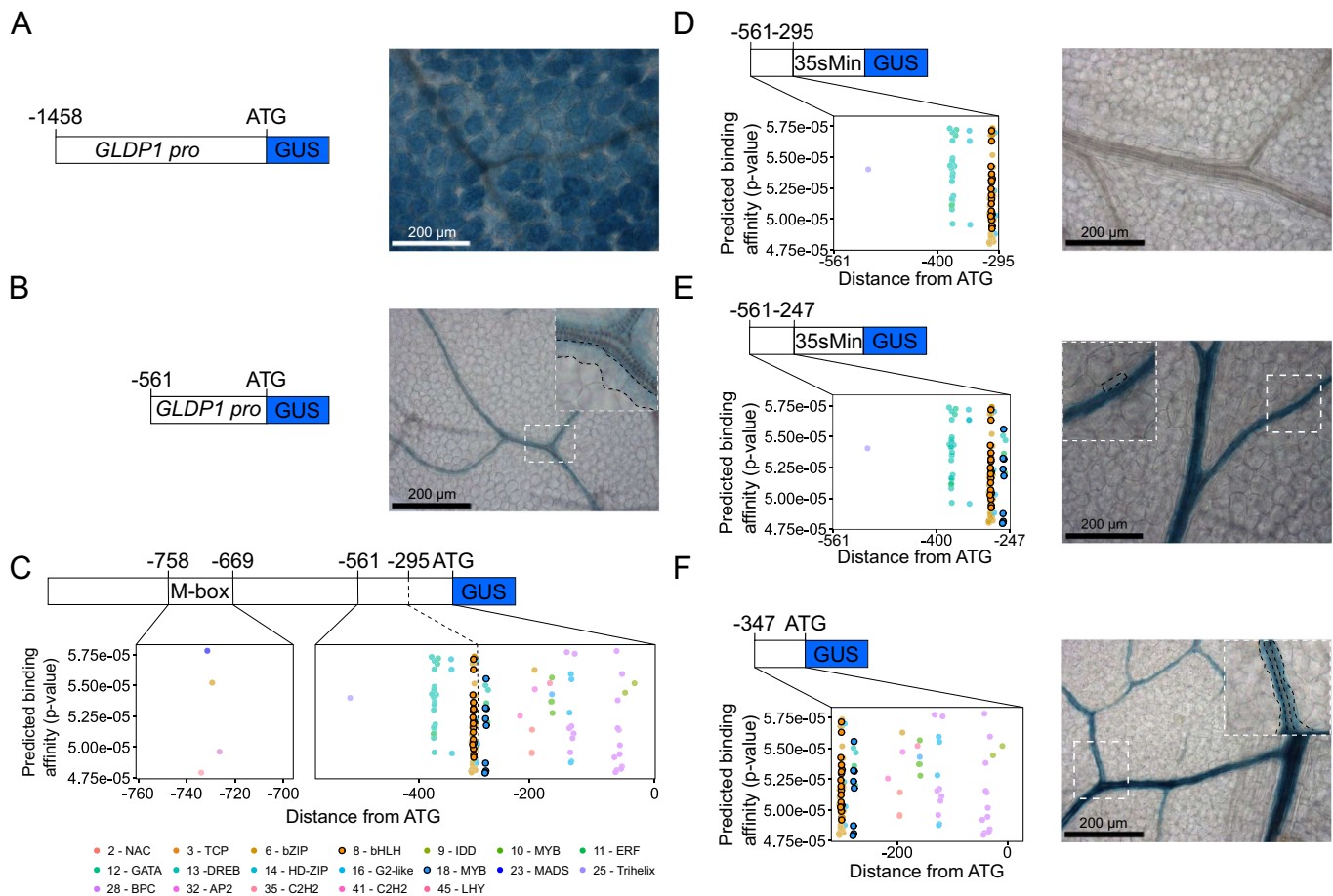

**Figure 1. MYC and MYB binding motifs control bundle sheath strand expression in *Arabidopsis thaliana*.**

(A) Schematic and representative GUS staining image of the full length—1458 bp *A. thaliana GLDP1* promoter upstream of the translational start site (ATG) from 19 independent T1 lines. (B) Schematic and representative GUS staining image of the -561 bp *A. thaliana GLDP1* promoter upstream of the ATG from 13 independent T1 lines (C) Predicted TF binding motifs in the M box and from −561 bp upstream to the ATG of the *A. thaliana GLDP1* promoter. The position in the promoter (bp) is on the *x* axis, and the predicted binding affinity (*P* values calculated from the log-likelihood score by the FIMO tool is on the *y* axis). The motifs are coloured by the motif clusters shown underneath the plots (Dataset EV1). (D–F) Transcription factor binding motifs and representative GUS staining images of nucleotides −561 to −295 bp upstream of the ATG fused to CaMV35smin (D), −561 to −247 bp upstream of the ATG fused to CaMV35smin (E) and −347 to the ATG. (F) from 17, 14 and 7 independent T1 lines, respectively. Distance from the ATG (bp) is on the *x* axis, and the predicted binding affinity (*P* values calculated from the log-likelihood score by the FIMO tool (Grant et al, 2011)) is on the *y* axis. On GUS images, leaves were stained for 24 h (A, B, E, F) or 48 h (D), scale bars are 200 μm and a zoomed in region of the image is marked by a dashed white box. Bundle sheath cells marked with dashed black line. Source data are available online for this figure.

## GLDP1 promoters from $C_2$ *Moricandia* species contain conserved and functional MYC and MYB binding sites

The Brassicaceae contains at least five independent origins of $C_2$ photosynthesis (Schlüter et al, 2023). We hypothesized that conservation of MYC and MYB binding sites driving bundle sheath expression of *GLDP1* across the Brassicaceae underpins repeated evolution of this trait. To test this, we aligned *GLDP1* promoter sequences from seventeen species across the Brassicaceae including nine $C_3$ and eight $C_2$ species representing the five independent origins of $C_2$ photosynthesis (Guerreiro et al, 2023). The MYC binding site (CACGTG) is perfectly conserved in all seventeen species analysed and the MYB binding site (CACCAAC) was perfectly conserved in fifteen of these seventeen species. The exceptions were $C_2$ *B. gravinae* and *D. tenuifolia* where a single substitution at position five of the motif replaced thymine with

adenine in the MYB binding site (Fig. 2A). This position is variable between thymine and adenine in TF binding motifs from the cluster of MYB TFs containing MYB28, MYB29 and MYB76 (Appendix Fig. S7). This suggests that the MYC and MYB binding sites responsible for driving bundle sheath strand expression of *GLDP1* in *A. thaliana* may be functional across these $C_3$ and $C_2$ Brassicaceae species. The data also indicate that *cis*-elements allowing expression in bundle sheath strands have remained stable for at least ~20.8 million years since the divergence of *Arabidopsis* and *Moricandia* (Schlüter et al, 2017).

To test whether these motifs are functional in Brassicaceae species in addition to *A. thaliana*, we used the *Moricandia* genus for further investigation. *Moricandia* contains $C_3$ as well as $C_2$ species and previous work indicated that a 561 nucleotide region containing these predicted MYC and MYB binding sites is necessary for expression in bundle sheath strands

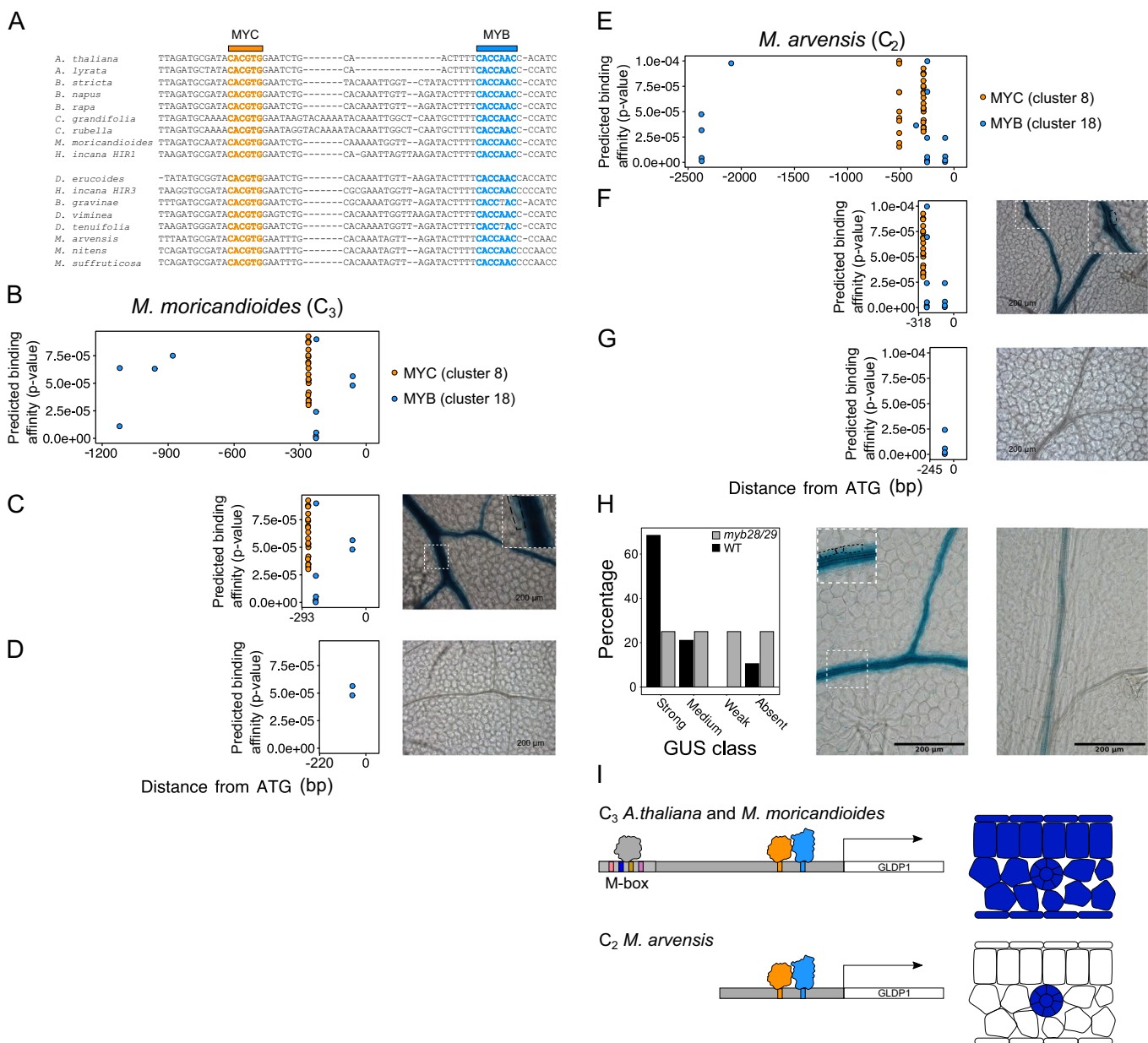

**Figure 2. MYC and MYB binding sites are conserved in the Brassicaceae and drive vein and bundle sheath preferential expression of *Moricandia GLDP1* genes.**

(**A**) Sequence alignments of the region of Brassicaceae *GLDP1* promoters containing MYC and MYB TF binding sites. MYC and MYB TF binding sites are coloured in gold and blue and marked above the alignment. (**B**) Position of MYC and MYB binding sites in the *M. moricandioides GLDP1* promoter. (**C**) Position of MYC and MYB binding sites and representative GUS staining images from 18 and 11 independent T1 lines, respectively, for *M. moricandioides* −293 bp and (**D**) −220 bp promoters. (**E**) Position of MYC and MYB binding sites in the *M. arvensis GLDP1* promoter. (**F**) Position of MYC and MYB binding sites and representative GUS staining images from 19 and 9 independent T1 lines, respectively, for *M. arvensis* −318 bp and (**G**) −245 bp promoters. Distance from the ATG (bp) is on the x axis, and the predicted binding affinity (*P* values calculated from the log-likelihood score by the FIMO tool (Grant et al, 2011) is on the y axis. On GUS images, leaves were stained for 24 h (**C**, **F**) or 48 h (**D**, **G**), scale bars are 200 μm and a zoomed in region of the image is marked by a dashed white box. Bundle sheath cells marked with dashed black line. (**H**) GUS staining of the *M. arvensis* 318 bp *GLDP1* promoter in the *myb28/29* mutant background. Classification of independent T1 lines into strong, medium, weak, or absent GUS activity for 12 lines in the *myb28/29* background and 19 in a WT background, distribution of GUS staining classes was significantly different between the two genotypes as determined by a chi-squared test (*P* value = 0.0377). Representative images of weak (top) and strong (bottom) GUS lines in a *myb28/29* mutant background. Leaves were stained for 24 h and scale bars are 200 μm. (**I**) Illustrative schematic showing model for the control of *GLDP1* expression in C₃ *A. thaliana* and *M. moricandioides* (top) and C₂ *M. arvensis* (bottom). In C₃ species constitutive expression is driven by unknown transcription factor(s) activating mesophyll expression from the M box (Adwy et al, 2015), potentially through binding to motifs from C2H2, MADS, bZIP and/or BPC families (Fig. 1C), and MYC and MYB TFs binding to closely spaced TF binding motifs to activate expression in the vein and bundle sheath. In C₂ species, the M box cannot activate expression in the mesophyll, but MYC and MYB binding sites are conserved leading to bundle sheath strand specific expression of *GLDP1*. Source data are available online for this figure.

(Adwy et al, 2015; Adwy et al, 2019). To test whether this patterning was due to these MYC and MYB binding sites we cloned fragments from *GLDP1* promoters of $C_3$ *M. moricandioides* and $C_2$ *M. arvensis*. In $C_3$ *M. moricandioides* closely spaced MYC and MYB sites are found between nucleotides -293 and -220 upstream of the predicted translational start site (Fig. 2B). Promoter deletions that removed all upstream sequence but retained or removed these motifs were generated. When the MYC and MYB sites were present GUS activity was detected in bundle sheath strands (Fig. 2C; Appendix Fig. S8) but when they were absent this was not the case (Fig. 2D; Appendix Fig. S9). Therefore, this $C_3$ member of *Moricandia* contains sequence in the *GLDP1* promoter that is recognised by the MYC-MYB module of *A. thaliana* and it is able to pattern gene expression to bundle sheath strands.

The *GLDP1* promoter from the $C_2$ species *M. arvensis* also has closely spaced MYC and MYB motifs (Fig. 2E). When they were present, GUS activity was detected in *A. thaliana* bundle sheath stands (Fig. 2F; Appendix Fig. S10) but when they were absent it was not (Fig. 2G; Appendix Fig. S11). These data define the MYC-MYB enhancer as being necessary and sufficient for the patterning of expression of *GLDP1* needed for $C_2$ photosynthesis, but do not confirm which transcription factors are responsible. To test this, we transformed the *A. thaliana myb28/29* mutant with the proximal enhancer allowing bundle sheath expression of $C_2$ *M. arvensis GLDP1*. Some expression of GUS in this mutant would be expected because MYB76 can also bind this motif and interact with MYC transcription factors to drive expression (Schweizer et al, 2013). We classified GUS staining into four classes from strong to absent. Notably, in controls more than 80% of lines showed strong or moderate expression in the bundle sheath, but when the enhancer was placed into the *myb28/29* mutant background this was significantly reduced (chi-squared test, *P* value = 0.037) (Fig. 2H; Appendix Fig. S12).

Taken together, these data show that the bipartite MYC and MYB transcription factor module responsible for directing *MYB76* and glucosinolate biosynthesis genes to bundle sheath strands of *A. thaliana* is also used to pattern expression of *GLDP1* to this tissue. Moreover, the bundle sheath enhancer containing a *cis*-code that is necessary and sufficient for bundle sheath strand expression is found in *GLDP1* genes from $C_3$ and $C_2$ species of *Moricandia*. The evolution of $C_2$ photosynthesis in the Brassicaceae is thus associated with retention of closely spaced MYC and MYB binding sites as well as a shift in position of the M box which disrupts its function (Triesch et al, 2024) such that *GLDP1* is expressed specifically in bundle sheath strands (Fig. 2I). In *Flaveria* although *trans*-factors have not been defined the changes in *cis* appear more complex. For example, two copies of *GLDP1* are expressed in $C_3$ species—one constitutively and one in the bundle sheath, and increased expression in the bundle sheath is mediated by both transcriptional and post-transcriptional mechanisms (Wiludda et al, 2012; Schulze et al, 2013).

In summary, our analysis reveals a molecular genetic mechanism underpinning the bundle sheath accumulation of glycine decarboxylase required for $C_2$ photosynthesis, and thus for a foundational step in the evolution of the $C_4$ photosynthetic pathway. Further analysis will be required to establish whether other $C_2$ and $C_4$ lineages have made use of this MYC-MYB transcription factor module or whether evolution has convergently recruited other transcription factor families to pattern genes to the bundle sheath.

# Methods

## Reagents and tools table

| Reagent/resource | Reference or source | Identifier or catalog number |
| --- | --- | --- |
| **Experimental models** | | |
| *Arabidopsis thaliana* Col-0 (WT) | Widely distributed | NASC id: N1093 |
| *Arabidopsis thaliana* myb28/myb29 | Gift from Professor Meike Burow (Burow et al, 2015) | N/A |
| **Recombinant DNA** | | |
| pICH41233 (P L0 acceptor) | Weber et al, 2011 | Addgene: 47984 |
| pICH41295 (PU L0 acceptor) | Weber et al, 2011 | Addgene: 47997 |
| pICH41246 (U L0 acceptor) | Weber et al, 2011 | Addgene: 47992 |
| pICH41308 (SC L0 acceptor) | Weber et al, 2011 | Addgene: 47998 |
| GUS reporter in SC acceptor | This study | N/A |
| FastR reporter in position L1R2 | This study, Shimada et al, 2010 | N/A |
| pICH41421 (Nos Terminator) | Engler et al, 2014 | Addgene: 50339 |
| pICH47811 (Level 1 acceptor position R2) | Weber et al, 2011 | Addgene: 48008 |
| pAGM4723 (L2 acceptor) | Weber et al, 2011 | Addgene: 48015 |
| *AtGLDP1pro1458bp_GUS_NosT_FastR_L2* | This study | N/A |
| *AtGLDP1pro561bp_GUS_NosT_FastR_L2* | This study | N/A |
| *AtGLDP1pro561to247bp_GUS_NosT_FastR_L2* | This study | N/A |
| *AtGLDP1pro347bp_GUS_NosT_FastR_L2* | This study | N/A |
| *MmGLDP1pro293bp_GUS_NosT_FastR_L2* | This study | N/A |
| *MmGLDP1pro220bp_GUS_NosT_FastR_L2* | This study | N/A |
| *MaGLDP1pro318bp_GUS_NosT_FastR_L2* | This study | N/A |
| *MaGLDP1pro245bp_GUS_NosT_FastR_L2* | This study | N/A |
| *CaMV35sMin* in U level 0 | This study | N/A |
| pJET1.2/blunt cloning vector | ThermoFisher | K1231 |
| **Antibodies** | | |
| N/A | | |
| **Oligonucleotides and other sequence-based reagents** | | |
| PCR primers | This study | Table EV2 |
| **Chemicals, enzymes and other reagents** | | |
| Bpi1 | ThermoFisher | ER1011 |
| Bsa1-HF v2 | New England Biolabs | R3733S |
| T4 DNA Ligase and buffer | ThermoFisher | EL0011 |
| BSA | ThermoFisher | B14 |
| Cla1 | New England Biolabs | R0197S |
| Xba1 | New England Biolabs | R0145S |
| NaOH | ThermoFisher | A16037.36 |
| Ethanol (96%) | Fisher Scientific | 15552393 |

| Reagent/resource | Reference or source | Identifier or catalog number |
|---|---|---|
| Acetic acid glacial | Fisher Scientific | 11475160 |
| Na$_2$HPO$_4$ | Fisher Scientific | 12665147 |
| NaH$_2$PO$_4$ | Fisher Scientific | 10227070 |
| Potassium ferricyanide | Merck | 702587 |
| Potassium ferrocyanide | Merck | ATEH99D1EC14 |
| EDTA | Merck | E9884 |
| Triton X-100 | Merck | T8787 |
| X-Gluc | Merck | AMBH9A9AF0DB |
| CloneJET PCR Cloning Kit | ThermoFisher | K1231 |
| Phusion PCR kit | ThermoFisher | F553L |
| DNeasy Plant Mini kit | Qiagen | 69104 |
| Zymoclean Gel DNA recovery kit | Zymo Research | D4002 |
| *Escherichia coli* DH5 alpha competent cells | Widely distributed | N/A |
| *Agrobacterium tumefaciens* GV3101 competent cells | Widely distributed | N/A |
| Spectinomycin | Merck | S4014 |
| Carbenicillin | Melford | C46000 |
| Kanamycin | Melford | K22000 |
| **Software** | | |
| FIMO | Meme suite (Grant et al, 2011) | https://meme-suite.org/meme/doc/fimo.html |
| Q Capture Pro 7 | QImaging | N/A |
| Rstudio, Version 4.3.2 | N/A | https://www.rstudio.com/ |
| RSAT | Castro-Mondragon et al, 2017 | https://rsat.eead.csic.es/plants/ |
| MUSCLE | Edgar, 2004 | N/A |
| UGENE | Okonechnikov et al, 2012 | N/A |
| JASPAR | Fornes, 2020 | https://jaspar.genereg.net |
| **Other** | | |
| Olympus BX41 light microscope | Olympus | N/A |
| QImaging MicroPublisher 3.3 RTV camera | QImaging | N/A |

## Plant materials and growth conditions

*A. thaliana* was grown on Levington F2 soil in growth chambers set at constant 20 °C, with a 16-h photoperiod with a light intensity of 150 µmol m$^{-2}$ s$^{-1}$ photon flux density, 60% relative humidity, and ambient CO$_2$ levels.

## Transcription factor binding site prediction and sequence alignments

Motif clustering was performed on plant transcription factor motifs downloaded from JASPAR using the RSAT tool (Castro-Mondragon et al, 2017) as reported previously (Dickinson et al, 2020). The FIMO tool (Grant et al, 2011) was used to scan DNA sequences for matches

to *A. thaliana* transcription factor binding motifs found in the JASPAR motif database (Fornes, 2020). To account for input sequence composition, a background model was generated using the fasta-get-markov tool from the MEME suite (Bailey et al, 2009). FIMO was then run with the default parameters and a *P* value cut-off of $1 \times 10^{-4}$.

Brassicaceae *GLDP1* promoter sequences were retrieved from phytozome (Goodstein et al, 2012) and promoters of *Moricandia* species were taken from Adwy et al (2019). Sequences were aligned using MUSCLE (Edgar, 2004) with default settings and alignments visualised with the UGENE tool (Okonechnikov et al, 2012).

## Cloning and GUS assays

Promoter GUS constructs were assembled using the Golden Gate system (Weber et al, 2011). Arabidopsis promoter fragments were isolated from genomic DNA by PCR (primers in Table EV1) and cloned into level 0 modules. *Moricandia* promoter fragments were initially amplified from genomic DNA using primers, adding a 5' ClaI and a 3' XbaI overhang. The amplified promoter sequences were subcloned into the pJET1.2 cloning vector using the Thermo Scientific CloneJET PCR Cloning Kit following the manufacturer's instructions. *Moricandia* promoter fragments for Golden Gate cloning were then amplified from these pJET vectors and cloning into level 0 modules. Level 1 constructs were then assembled to fuse the promoter fragments with the *CaMV 35sMinimal* promoter were required, the GUS reporter and Nos terminator. Level 2 constructs were then assembled to add the FastR selectable marker (Shimada et al, 2010) to allow selection of positive transformants. Level 2 constructs were then placed into *Agrobacterium tumefaciens* strain GV3101 and introduced into *A. thaliana* Col-0 by floral dipping (Clough and Bent, 1998).

To take into account position effects associated with the transgene insertion site, GUS staining was undertaken on at least six randomly selected T1 plants for each *uidA* fusion. The staining solution contained 0.1 M Na$_2$HPO$_4$ (pH 7.0), 2 mM potassium ferricyanide, 2 mM potassium ferrocyanide, 10 mM EDTA (pH 8.0), 0.06% (v/v) Triton X-100 and 0.5 mg ml$^{-1}$ X-gluc. Leaves from three-week-old plants were vacuum-infiltrated three times in GUS solution for one minute and then incubated at 37 °C for 24 h. Next, stained samples were fixed in 3:1 (v/v) ethanol:acetic acid for 30 min at room temperature, cleared in 70% (v/v) ethanol at 37 °C and then placed in 5 M NaOH for 2 h. The samples were stored in 70% (v/v) ethanol at 4 °C. The samples were imaged with an Olympus BX41 light microscope with Q Capture Pro 7 software and a QImaging MicroPublisher 3.3 RTV camera.

# Data availability

This study includes no data deposited in external repositories.

The source data of this paper are collected in the following database record: biostudies:S-SCDT-10_1038-S44319-025-00461-1.

# Peer review information

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

## Acknowledgements

The work was funded by and European Research Council Advanced Grant 694733 REVOLUTION, BBSRC grant BBW00013X1 to JMH and European Union Program (project GAIND4CROPS GA number 862087) to JMH and APMW. Work in the group of APMW was funded by ERA-CAPS project "C4BREED" under Project ID WE 2231/20–1, the Cluster of Excellence for Plant Sciences (CEPLAS) under Germany's Excellence Strategy EXC-2048/1 under project ID 390686111, and the CRC TRR 341 "Plant Ecological Genetics" grant by the German Research Foundation (DFG). For the purpose of open access, the authors have applied a Creative Commons Attribution (CC BY) license to any Author Accepted Manuscript version arising from this submission.

## Author contributions

**Patrick J Dickinson**: Conceptualization; Resources; Data curation; Formal analysis; Investigation; Methodology; Writing—original draft; Writing—review and editing. **Sebastian Triesch**: Resources; Data curation; Formal analysis; Investigation; Writing—original draft. **Urte Schlüter**: Formal analysis; Supervision; Methodology; Writing—review and editing. **Andreas P M Weber**: Conceptualization; Supervision; Funding acquisition; Writing—original draft; Writing—review and editing. **Julian M Hibberd**: Conceptualization; Supervision; Funding acquisition; Investigation; Writing—original draft; Project administration; Writing—review and editing.

Source data underlying figure panels in this paper may have individual authorship assigned. Where available, figure panel/source data authorship is listed in the following database record: biostudies:S-SCDT-10_1038-S44319-025-00461-1.

## Disclosure and competing interests statement

The authors declare no competing interests.

