## [Peer Review File · EMBO Reports]

A transcription factor module mediating C2 photosynthesis in the Brassicaceae

Julian Hibberd, Patrick Dickinson, Sebastian Triesch, Urte Schlüter, and Andreas Weber

Corresponding author(s): Julian Hibberd (jmh65@cam.ac.uk) , Patrick Dickinson (pd373@cam.ac.uk)

Review Timeline:

Submission Date:	17th Sep 24
Editorial Decision:	15th Nov 24
Appeal Received:	27th Nov 24
Editorial Decision:	23rd Dec 24
Appeal Received:	17th Jan 25
Editorial Decision:	11th Mar 25
Revision Received:	14th Mar 25
Accepted:	7th Apr 25

Transaction Report:

Dear Dr. Hibberd

Thank you for the submission of your manuscript to EMBO reports. Please accept my sincere apologies regarding the delay in handling your manuscript. We have received the enclosed reports on it.

I am sorry to say that the evaluation of your manuscript is not a positive one. As you will see, the referees acknowledge the value of your findings for the more immediate community, but also note that the data are to some extent confirmatory and limited to Brassicaceae.

Given these concerns regarding the broader conceptual advance and novelty of the findings and the fact that EMBO reports can only invite revision of papers that receive enthusiastic support from a majority of referees, I am sorry to say that we cannot offer to publish your manuscript.

While we cannot offer publication of your manuscript, I feel that your study is an excellent candidate for our partner journal Life Science Alliance (<http://www.life-science-alliance.org/>) that takes referee reports from EMBO Press into account when reaching a decision. In case you are interested in discussing a potential revision of your study at Life Science Alliance, please contact Executive Editor Eric Sawey (e.sawey@life-science-alliance.org), who will be pleased to answer any questions.

I am sorry to disappoint you on this occasion, and hope that the referee comments will be helpful in your continued work in this area.

Yours sincerely

=====

Referee #1:

This is an interesting study that sought to understand the mechanisms contributing to bundle sheath-specific expression of GDC in plants using C2 photosynthesis. First, they show that the joint mesophyll and bundle sheath expression of GDC (specifically, AtGLDP1) found in C3 plants is achieved via a combined effect of multiple elements, such as the M box and MYC-MYB modules; and that MYC-MYB can function alone to maintain bundle sheath expression if other contributing elements are made ineffectual. Next, they found that the MYC binding sites are conserved across all C2 species (and all independent origins of C2) in the Brassicaceae, while the MYB sites were entirely conserved across 6 of the 8 C2 Brassicaceae species studied. This work will be of interest to the C2 and C4 engineering communities, as well as those interested in photosynthetic diversity evolution more broadly. Minor suggestions below.

L65-66. A pedantic point but technically, C3-C4 and C2 are not interchangeable terms. Instead, C2 is a subset of C3-C4 intermediates, specifically those with an active M-BS glycine shuttle (i.e., and not, for example, proto-kranz C3-C4 intermediates).

L70. Suggest "and then a shift of glycine decarboxylase [AWAY] from mesophyll cells such that its activity is restricted to the bundle sheath"

L72a. "Repositioning of glycine decarboxylase to the bundle sheath is conjectured to initiate greater rates of CO₂ release" [Suggest cite Keerberg et al. 2014. C2 photosynthesis generates about 3-fold elevated leaf CO₂ levels in the C3-C4 intermediate species *Flaveria pubescens*. *Journal of Experimental Botany* 65: 3649-3656.]

L73-75. "The two-carbon glycine molecule thus provides CO₂ for photosynthesis [in bundle sheath tissue] and[, as such,] led to the term C2 photosynthesis.

Figure 2 Panel I. Do you have images of the GUS staining in cross section to include in addition to (or instead of) the drawn diagram.

L232. Were the chambers set to 20C temp over the day and night? And was CO₂ controlled or ambient?

Referee #2:

The expression of Glycine decarboxylase P-subunit (GLDP) in the bundle sheath cell is a crucial characteristic of C2 photosynthesis. In their manuscript, Dickinson et al. aimed to understand the transcription factors that controls the expression of Glycine decarboxylase in the dundle sheath cell. They focused on the mechanism underlying the expression of GLDP in the bundle sheath, particularly on GLDP1 as C2 species of the Brassicaceae lineages have lost GLDP2 and have only GLDP1. To achieve this, the authors first identified closely spaced MYC and MYB binding sites in C3 Arabidopsis species that regulate the expression of GLDP1 to bundle sheath cells using the promoter GUS. They then checked if this module was conserved in other species by aligning the GLDP1 promoter sequences from nine C3 species and eight C2 species. They found that it was conserved in all species except two C2 species. They also tested if the MYC-MYB motifs were functional in C3 and C2 species of Moricandia by cloning the MYC-MYB binding sites of these species. This study provides valuable insights for the community working on the evolution of C4 photosynthesis. However, the authors' findings are specific to Brassicaceae due to the limited sample size used in the study. Consequently, the title and related conclusions should be modified accordingly.

1. Does this manuscript report a single key finding? YES/NO

YES, the closely spaced MYC and MYB binding sites control the expression of glycine decarboxylase, which is required for C2 photosynthesis.

2. Is the reported work of significance (YES), or does it describe a confirmatory finding or one that has already been documented using other methods or in other organisms etc (NO)?

No, it describes a confirmatory finding that has already been documented using similar species (<https://onlinelibrary.wiley.com/doi/epdf/10.1111/tpj.13084>).

3. Is it of general interest to the molecular biology community? YES/NO

Yes

4. Is the single major finding robustly documented using independent lines of experimental evidence (YES), or is it really just a preliminary report requiring significant further data to become convincing, and thus more suited to a longerÅ-format article (NO)?

Yes

A.Minor comments

1. In the text on lines 58-59, you were discussing CO2 fixation. However, you suddenly mentioned the fixation of bicarbonate, which disrupted the flow and made it difficult to understand because there was no link with the previous sentence.

2. With the picture (Fig. 1A), it is not possible to differentiate between BSC and MC. Therefore, it cannot support the authors' claimed.

3. There is no Supplementary Table 1.

4. Supplemental Figure 2 Legends are missing, making it difficult to understand the pictures. Additionally, a clear indication of BSC and Mc would make comprehension easier.

5. Lines 135-139. "We thus consider this smallchange in GLDP1 expression in the myb28/29 double mutant consistent with MYB transcription factors controlling GLDP1 expression in the bundle sheath." This statement is speculative without evidence. It should be reformulated unless a correlation between the cell size and the expression level is provided.

B. Major comments

1. Why are the authors using previously published data without conducting their own research to confirm?

2. Lines 176-177, In two C2 species, *B. gravinae* and *D. tenuifolia*, the binding sites of MYC and MYB were not found. Out of 8 C2 species, these binding sites were conserved in only 2 species, which is 25%, and not an exception. The conclusion drawn is speculative.

** As a service to authors, EMBO Press provides authors with the ability to transfer a manuscript that one journal cannot offer to publish to another journal, without the author having to upload the manuscript data again. To transfer your manuscript to another EMBO Press journal using this service, please click on

Link Not Available

Dear Martina,

Thank you for handling the manuscript. We understand your decision based on the text you have been provided from the assessment of Reviewer 2. Their main point of criticism appears to be that it is confirmatory, but this is not correct. Apart from this, their assessment appears rather positive overall, *i.e.* has no real specific criticism of the quality of the work.

The points leading to the decline decision because 'the work is confirmatory, and of limited interest' are addressed below. We note that they comment on the fact the data are derived from the Brassicaceae, which is true, because it is the first time such a transcription factor system has been identified. Decades of work on models such as Arabidopsis, Drosophila, mouse etc validated this approach, and it is not feasible to translate such a finding into the more than sixty lineages of plants that use C₄ photosynthesis anyway as most are not transformable. (The comments from Reviewer 1 are reasonable and we can address them.)

Referee #2:

2. Is the reported work of significance (YES), or does it describe a confirmatory finding or one that has already been documented using other methods or in other organisms etc (NO)?

No, it describes a confirmatory finding that has already been documented using similar species (<https://onlinelibrary.wiley.com/doi/epdf/10.1111/tpj.13084>).

*The reviewer has made an error - either they did not carefully read the work of Adwy et al. that they provide a link for, or they misunderstood it. Adwy et al. identified **a mesophyll box and a 266bp vein/bundle sheath box** (see their paper) controlling GLDP1 expression in the mesophyll and vein/bundle sheath respectively. Neither the cis-elements nor the transcription factors involved in generating either mesophyll or bundle sheath expression were identified and this was what stimulated our analysis.*

Our analysis identifies for the first time the transcription factors and actual cis-elements that they bind that control bundle sheath expression. This is not confirmatory, it is novel and a conceptual advance as it identifies the molecular players allowing the evolution of C₂ photosynthesis.

(Please also note that the Adwy work on the mesophyll box has subsequently been updated to show that the mesophyll box has not been lost as they proposed, but rather a transposon insertion moved it (<https://doi.org/10.1111/plb.13601>).

B. Major comments

1. Why are the authors using previously published data without conducting their own research to confirm?

This is not correct. All of our conclusions are based on our own research (every panel in every figure is from our lab. And, of all panels in the S Figures, (total 11), only S Figure 1 is a

reanalysis of publicly available data, and we acknowledge this in the legend. We therefore find this assessment incorrect and worrisome.

2. Lines 176-177, In two C2 species, *B. gravinae* and *D. tenuifolia*, the binding sites of MYC and MYB were not found. Out of 8 C2 species, these binding sites were conserved in only 2 species, which is 25%, and not an exception. The conclusion drawn is speculative.

Again, this is not correct – and to a worrying extent. Of these eight species the binding sites were absolutely conserved in 6 out of 8 species (75%) - not 2 out of 8 (25%) as they state.

*Moreover, their broader summary of this section is flawed. The MYC and MYB binding sites are found in both C₂ species that they commented on. We pointed out in the manuscript that in those two species we detected a single base pair substitution. So, by looking at the nine C₃ and eight C₂ species we found that **across all seventeen** species the binding sites were **conserved**, and in **fifteen they were identical**.*

In addition, as demonstrated experimentally over decades, a single base pair change in a motif bound by transcription factors may reduce binding but is very unlikely to abolish it. In the case of the specific motif here, the site with the point mutation is the least conserved, and is documented to allow binding by MYBs.

We are not sure how you would like to proceed, but would value your thoughts on the above.

Best wishes,

Julian Hibberd on behalf of the authors.

Dear Dr. Hibberd

Thank you for your letter asking us to reconsider our decision and invite revision of your manuscript. I apologize for my delayed response, but I have now carefully read your letter and re-read the referee reports.

I analysed the comments from referee #2 and your response to these and largely agree with your line of argumentation. I also note that referee #1 was more positive about your work, appreciating its potential interest. I am therefore in principle open to consider a re-submission of your work that addresses all concerns in full, which would be evaluated again by referee #1, complemented by an advisor. If you agree with this proposal, please address all referee concerns and submit your manuscript *de novo*. Please note that the manuscript will be treated as a new submission in terms of novelty assessment at the time of submission, but as outlined above, it will be seen by the same referee and thus handled like a revised manuscript.

Please do not hesitate to contact me in case you have any questions or would like to discuss the procedure further.

Kind regards,

** As a service to authors, EMBO Press provides authors with the ability to transfer a manuscript that one journal cannot offer to publish to another journal, without the author having to upload the manuscript data again. To transfer your manuscript to another EMBO Press journal using this service, please click on Link Not Available

Dear Martina,

re. Previous manuscript "EMBOR-2024-60401V2-Q".

Thank you very much for your feedback, and the proposal in your email from 23/12/2024 that we respond to comments from both referees, such that referee 1 and an advisor can assess the changes that we have made. Below we summarise our responses.

Referee #1:

This is an interesting study that sought to understand the mechanisms contributing to bundle sheath-specific expression of GDC in plants using C2 photosynthesis. First, they show that the joint mesophyll and bundle sheath expression of GDC (specifically, AtGLDP1) found in C3 plants is achieved via a combined effect of multiple elements, such as the M box and MYC-MYB modules; and that MYC-MYB can function alone to maintain bundle sheath expression if other contributing elements are made ineffectual. Next, they found that the MYC binding sites are conserved across all C2 species (and all independent origins of C2) in the Brassicaceae, while the MYB sites were entirely conserved across 6 of the 8 C2 Brassicaceae species studied. This work will be of interest to the C2 and C4 engineering communities, as well as those interested in photosynthetic diversity evolution more broadly. Minor suggestions below.

Response: Thank you for this careful and positive assessment of the findings, we have addressed each of the minor suggestions below.

L65-66. A pedantic point but technically, C3-C4 and C2 are not interchangeable terms. Instead, C2 is a subset of C3-C4 intermediates, specifically those with an active M-BS glycine shuttle (i.e., and not, for example, proto-kranz C3-C4 intermediates).

Response: We agree, and apologise for the lack of clarity in our initial writing. We have modified the text to ensure accuracy, and it now reads "Such plants are known as C₃-C₄ intermediates and a subset of these with an active mesophyll to bundle sheath glycine shuttle are known as C₂ species" (please see lines 65-67).

L70. Suggest "and then a shift of glycine decarboxylase [AWAY] from mesophyll cells such that its activity is restricted to the bundle sheath"

Response: Thank you, we agree this is an improved phrasing, and we have included 'away' in the sentence (it now reads "and then a shift of glycine decarboxylase away from mesophyll cells such that its activity is restricted to the bundle sheath" please see line 70).

L72a. "Repositioning of glycine decarboxylase to the bundle sheath is conjectured to initiate greater rates of CO₂ release" [Suggest cite Keerberg et al. 2014. C2 photosynthesis generates about 3-fold elevated leaf CO₂ levels in the C3-C4 intermediate species *Flaveria pubescens*. Journal of Experimental Botany 65: 3649-3656.]

Response: This is a good point, and we have now cited this work (lines 74-75).

L73-75. "The two-carbon glycine molecule thus provides CO₂ for photosynthesis [in bundle sheath tissue] and[, as such,] led to the term C2 photosynthesis.

Response: Thank you for the suggestion, we agree and the text now reads “The two-carbon glycine molecule thus provides CO₂ for photosynthesis in bundle sheath tissue and, as such, led to the term C₂ photosynthesis.” (please see lines 74-75)

Figure 2 Panel I. Do you have images of the GUS staining in cross section to include in addition to (or instead of) the drawn diagram.

Response: We apologise for the lack of clarity in our purpose here. Originally, we were using this diagram as a summary model of the GUS staining presented in the rest of the paper and it is meant to be illustrative rather than more data. We have modified the figure legend (line 464) so that it now reads “Schematic of the model we propose for the control of GLDP1 expression” to ensure it is clear that these are not meant to be new data, but rather a model summarizing our findings.

L232. Were the chambers set to 20C temp over the day and night? And was CO₂ controlled or ambient?

Response: We apologise that this information was not present. We have now modified the text (lines 238-240) so that it is clear that temperature was set to be a constant 20°C over the day and night, and that CO₂ was ambient.

Referee #2:

The expression of Glycine decarboxylase P-subunit (GLDP) in the bundle sheath cell is a crucial characteristic of C₂ photosynthesis. In their manuscript, Dickinson et al. aimed to understand the transcription factors that controls the expression of Glycine decarboxylase in the dundle sheath cell. They focused on the mechanism underlying the expression of GLDP in the bundle sheath, particularly on GLDP1 as C₂ species of the Brassicaceae lineages have lost GLDP2 and have only GLDP1. To achieve this, the authors first identified closely spaced MYC and MYB binding sites in C₃ Arabidopsis species that regulate the expression of GLDP1 to bundle sheath cells using the promoter GUS. They then checked if this module was conserved in other species by aligning the GLDP1 promoter sequences from nine C₃ species and eight C₂ species. They found that it was conserved in all species except two C₂ species. They also tested if the MYC-MYB motifs were functional in C₃ and C₂ species of Moricandia by cloning the MYC-MYB binding sites of these species. This study provides valuable insights for the community working on the evolution of C₄ photosynthesis. However, the authors' findings are specific to Brassicaceae due to the limited sample size used in the study. Consequently, the title and related conclusions should be modified accordingly.

Response: Thank you for the summary. Along with a very large community, we did indeed use the Brassicaceae to understand the biological question being addressed. To reflect this in the title, we have added “in the Brassicaceae” to the title.

1. Does this manuscript report a single key finding? YES/NO

YES, the closely spaced MYC and MYB binding sites control the expression of glycine decarboxylase, which is required for C₂ photosynthesis.

Response: Thank you for this assessment.

2. Is the reported work of significance (YES), or does it describe a confirmatory finding or one that has already been documented using other methods or in other organisms etc (NO)?

No, it describes a confirmatory finding that has already been documented using similar species (<https://onlinelibrary.wiley.com/doi/epdf/10.1111/tpj.13084>).

Response: *We believe that the reviewer has made an error here. Adwy et al. identified a mesophyll box and also a 266 base pair vein/bundle sheath box (see hyper-linked paper above) controlling GLDP1 expression in the mesophyll and vein/bundle sheath respectively. Within the mesophyll box and the 266 base pair region reported by Adwy et al., neither the cis-elements nor the transcription factors involved in generating either mesophyll or bundle sheath expression were identified, and this was what stimulated our analysis.*

In contrast, the data we report here identify for the first time the transcription factors and actual cis-elements that they bind to generate bundle sheath expression. Our data are therefore not confirmatory. In fact, the data represent a conceptual advance because they identify the first molecular players allowing the evolution of C₂ photosynthesis.

(Please also note that the Adwy work on the mesophyll box has subsequently been updated to show that the mesophyll box has not been lost as they proposed, but rather a transposon insertion moved it (<https://doi.org/10.1111/plb.13601>).

3. Is it of general interest to the molecular biology community? YES/NO

Yes.

Response: *Thank you for this assessment.*

4. Is the single major finding robustly documented using independent lines of experimental evidence (YES), or is it really just a preliminary report requiring significant further data to become convincing, and thus more suited to a longer format article (NO)?

Yes

Response: *Thank you for this assessment.*

A. Minor comments

1. In the text on lines 58-59, you were discussing CO₂ fixation. However, you suddenly mentioned the fixation of bicarbonate, which disrupted the flow and made it difficult to understand because there was no link with the previous sentence.

Response: *Thank you for raising this point. It was because PEPC does not fix CO₂ but rather only interacts with HCO₃. We have now modified the text such that it now reads "Typically, after conversion of CO₂ to bicarbonate by carbonic anhydrases C₄ biochemistry enables fixation by the enzyme phosphoenolpyruvate carboxylase in mesophyll cells" (now on lines 58-59).*

2. With the picture (Fig. 1A), it is not possible to differentiate between BSC and MC. Therefore, it cannot support the authors' claimed.

Response: Thank you for this point. We agree, it is difficult to see both cell types, but this is not easy to do in one image. Here we were primarily aiming to use this image to show that the full-length promoter drives strong expression in mesophyll cells. As this sort of imaging does not allow high quality images with both cell types to be in focus, we have also provided images in Supplemental Fig 2 where staining of bundle sheath cells is clear. To clarify this, we have modified the text on lines 112-113 to read “with expression in mesophyll cells (Fig. 1A, Supplemental Fig. 2) and bundle sheath strands (Supplemental Fig. 2)”.

3. There is no Supplementary Table 1.

Response: Supplementary Table 1 is included as a separate spreadsheet that is distinct from the figures and contains the list of transcription factors and their clustering.

4. Supplemental Figure 2 Legends are missing, making it difficult to understand the pictures. Additionally, a clear indication of BSC and Mc would make comprehension easier.

Response: The legends are/were present in Supplemental Figure 2 (eg see page 3). We are not sure why the referee was unable to see them. To clarify we have indicated mesophyll cells and bundle sheath strands on the first panel of Supplemental figure 2.

5. Lines 135-139. "We thus consider this small change in GLDP1 expression in the myb28/29 double mutant consistent with MYB transcription factors controlling GLDP1 expression in the bundle sheath." This statement is speculative without evidence. It should be reformulated unless a correlation between the cell size and the expression level is provided.

Response: There is no mention of cell size in the text and so this is not relevant. We state that this small reduction in expression is consistent with MYB control of expression because the bundle sheath makes up only 10-15% of cells in the leaf. Therefore, as MYBs do not affect the majority of leaf cells, a small reduction would be expected when their function is perturbed. We therefore consider this statement accurate and have kept it in (lines 140-142).

B. Major comments

1. Why are the authors using previously published data without conducting their own research to confirm?

Response: All of our conclusions are based on our own research (every panel in every figure is from our lab). Of the panels in the Supplementary Figures (totalling eleven), only Supplementary Figure 1 is a reanalysis of publicly available data, and we acknowledge this in the legend. We therefore find this assessment incorrect.

2. Lines 176-177, In two C2 species, *B. gravinae* and *D. tenuifolia*, the binding sites of MYC and MYB were not found. Out of 8 C2 species, these binding sites were conserved in only 2 species, which is 25%, and not an exception. The conclusion drawn is speculative.

Response: Again, this is not correct. In the eight species we assessed, the binding sites were absolutely conserved in six out of eight species (75%) - not two out of eight (25%) as the referee writes.

Moreover, their broader summary of this section is flawed. The MYC and MYB binding sites are found in both C₂ species (*B. gravinae* and *D. tenuifolia*) that they commented on. We pointed out in the manuscript that in those two species we detected a single base pair substitution. So, by looking collectively at the nine C₃ and eight C₂ species, we found that across all seventeen species the binding sites were conserved, and in fifteen they were identical. As demonstrated experimentally over decades, a single base pair change in a motif bound by transcription factors may reduce binding but is very unlikely to abolish it. In the case of the specific motif here, the site with the point mutation is variable in MYB binding sites from this family. To make this point clearer we have modified the text to read as follows:

“The MYC binding site (CACGTG) is perfectly conserved in all seventeen species analysed and the MYB binding site (CACCAAC) was perfectly conserved in fifteen of these seventeen species. The exceptions were C₂ *B. gravinae* and *D. tenuifolia* where a single substitution at position five of the motif replaced thymine with adenine in the MYB binding site. This position is variable between thymine and adenine in transcription factor binding motifs from the cluster of MYBs containing MYB28, MYB29 and MYB76 (Supplemental Fig. 7).” (please see lines 177-183).

We have also added a supplemental figure showing TF binding motifs from the cluster of MYB TFs containing MYB28, MYB29 and MYB76 and modified the text to read: “This position is variable between thymine and adenine in transcription factor binding motifs from the cluster of MYB TFs containing MYB28, MYB29 and MYB76 (Supplemental Fig. 7)”.

Dear Dr. Hibberd

Thank you for the submission of your revised manuscript to our journal. It has been seen again by former referee 1 who considers your response, also to the concerns raised by referee 2, adequate and supports publication.

Before we can proceed with the official acceptance, I kindly ask you to format your article and provide further files according to our editorial policies. Once the revised manuscript is resubmitted, our data editors and editorial assistants will perform a few checks (source data, statistics, figure legends etc) and once this is done you will receive and accept-in-principle letter detailing some last minor changes that might be required.

You will find general formatting guidelines below, but in order to make the process smooth and faster, I list a few specific points here:

- 1) Supplemental table S1 is a datasets Please upload it as file type "Dataset". The nomenclature is Dataset EV1. The legend should be provided in a separate tab of the file. Please also add the header Dataset EV1 to the legend.
- 2) Supplemental table S2 can be incorporated in the Reagents and Tools table. Alternatively, it should be call either Table EV1 or Table 1.
- 3) Supplemental Figures 1 - 12 can remain in the format of a single PDF but the nomenclature should be changed to Appendix Fig S# . The PDF is called Appendix and needs a title page with a table of content and page numbers.
- 4) Triesch et al is a preprint. Please cite it in the text as (preprint: Triesch et al, 2022) and add [PREPRINT] to the citation in the reference list. Please also provide the DOI.
- 5) Funding information in the Acknowledgment and in the online manuscript tracking system must match. It seems that the following grants have not been entered in the system:
The Cluster of Excellence for Plant Sciences (CEPLAS) under Germany's Excellence Strategy EXC-2048/1 under project ID 390686111, and the CRC TRR 341 "Plant Ecological Genetics" grant by the German Research Foundation (DFG).
- 6) Please define the author contributions in the online system. This information will be automatically retrieved and typeset into the article.
- 7) References: please use et al if there are more than 10 authors.
- 8) Materials and Methods are called methods and we require a Reagents and Tools table (see point 12 below).
- 9) We need the Author checklist (point 4 below).
- 10) There are two author name discrepancies: Dr. Urte Schlüter is listed as "Dr. Urte Schlüter Urte Schlüter" in the online submission system.
Andreas Weber is "Andreas P.M. Weber" on the paper and "Andreas P. Weber" in the system.
- 11) Please provide source data (minimally modified data used to generate graphs and figures. We need one folder per main figure with subfolders for each panel. Source data for Appendix figures is not mandatory.

GENERAL formatting guidelines:

- 1) a .docx formatted version of the manuscript text (including legends for main figures, EV figures and tables). Please make sure that the changes are highlighted to be clearly visible.
- 2) individual production quality figure files as .eps, .tif, .jpg (one file per figure).
Please download our Figure Preparation Guidelines (figure preparation pdf) from our Author Guidelines pages <https://www.embopress.org/page/journal/14693178/authorguide> for more info on how to prepare your figures.
- 3) a .docx formatted letter INCLUDING the reviewers' reports and your detailed point-by-point responses to their comments. As part of the EMBO Press transparent editorial process, the point-by-point response is part of the Review Process File (RPF), which will be published alongside your paper.
- 4) a complete author checklist, which you can download from our author guidelines

(<<https://www.embopress.org/page/journal/14693178/authorguide>>). Please insert information in the checklist that is also reflected in the manuscript. The completed author checklist will also be part of the RPF.

5) Please note that all corresponding authors are required to supply an ORCID ID for their name upon submission of a revised manuscript (<<https://orcid.org/>>). Please find instructions on how to link your ORCID ID to your account in our manuscript tracking system in our Author guidelines (<<https://www.embopress.org/page/journal/14693178/authorguide#authorshipguidelines>>)

6) We replaced Supplementary Information with Expanded View (EV) Figures and Tables that are collapsible/expandable online. A maximum of 5 EV Figures can be typeset. EV Figures should be cited as "Figure EV1, Figure EV2" etc... in the text and their respective legends should be included in the main text after the legends of regular figures.

- For the figures that you do NOT wish to display as Expanded View figures, they should be bundled together with their legends in a single PDF file called *ure*, which should start with a short Table of Content. Appendix figures should be referred to in the main text as: "Appendix Figure S1, Appendix Figure S2" etc. See detailed instructions regarding expanded view here: <<https://www.embopress.org/page/journal/14693178/authorguide#expandedview>>

7) Please include a dedicated "Data Availability" section at the end of the Methods (suggested wording: "The [structural coordinates | microarray | mass spectrometry] data from this publication have been deposited to the [name of the database] database [URL] and assigned the identifier [accession | permalink | hashtag]."). Should this not apply, this should still be stated as "This study includes no data deposited in external repositories."

Additional information on source data and instruction on how to label the files are available <<https://www.embopress.org/page/journal/14693178/authorguide#sourcedata>>.

10) Figure legends and data quantification:
The following points must be specified in each figure legend:

- the name of the statistical test used to generate error bars and P values,
- the number (n) of independent experiments (please specify technical or biological replicates) underlying each data point,
- the nature of the bars and error bars (s.d., s.e.m.)
- If the data are obtained from n {less than or equal to} 5, show the individual data points in addition to the SD or SEM.
- If the data are obtained from n {less than or equal to} 2, use scatter blots showing the individual data points.

See also the guidelines for figure legend preparation:
<https://www.embopress.org/page/journal/14693178/authorguide#figureformat>

11) Our journal encourages inclusion of *data citations in the reference list* to directly cite datasets that were re-used and obtained from public databases. Data citations in the article text are distinct from normal bibliographical citations and should directly link to the database records from which the data can be accessed. In the main text, data citations are formatted as follows: "Data ref: Smith et al, 2001" or "Data ref: NCBI Sequence Read Archive PRJNA342805, 2017". In the Reference list, data citations must be labeled with "[DATASET]". A data reference must provide the database name, accession number/identifiers and a resolvable link to the landing page from which the data can be accessed at the end of the reference. Further instructions are available at <<https://www.embopress.org/page/journal/14693178/authorguide#referencesformat>>.

12) All Materials and Methods need to be described in the main text using our 'Structured Methods' format. According to this format, the Methods section includes a Reagents and Tools Table (listing key reagents, experimental models, software and relevant equipment and including their sources and relevant identifiers) followed by a Methods and Protocols section describing the methods, ideally using a step-by-step protocol format. The aim is to facilitate adoption of the methodologies across labs. Please download and fill our Reagents and Tools Table template (.docx), which you can find in our author guidelines: <https://www.embopress.org/page/journal/14693178/authorguide#structuredmethods>.

An example of a Method paper with Structured Methods can be found here: <https://www.embopress.org/doi/10.15252/msb.20178071>.

13) As part of the EMBO publication's Transparent Editorial Process, EMBO Reports publishes online a Review Process File to accompany accepted manuscripts. This File will be published in conjunction with your paper and will include the referee reports, your point-by-point response and all pertinent correspondence relating to the manuscript.

Kind regards,

=====

Referee #1:

The authors do a good job addressing points from both reviewers in this revision. I believe that the authors are correct in highlighting that Reviewer 2 missed the point (and, therefore, the novelty) of the study. Not only is this submission NOT redundant to the Adwy et al. work, it is indeed a distinct and important advancement in the field and worthy of publication in EMBO, in my opinion.

All editorial and formatting issues were resolved by the authors.

Julian Hibberd
University of Cambridge
Department of Plant Sciences
Downing Street
Cambridge CB2 3EA
United Kingdom

Dear Dr. Hibberd,

I am very pleased to accept your manuscript for publication in the next available issue of EMBO reports. Thank you for your contribution to our journal.

Yours sincerely,
